# Invertible Tabular GANs: Killing Two Birds with One Stone for Tabular Data Synthesis

**Jaehoon Lee**
Yonsei University
ljh5694@yonsei.ac.kr

**Jihyeon Hyeong**
Yonsei University
jiji.hyeong@yonsei.ac.kr

**Jinsung Jeon**
Yonsei University
Jjsjjs0902@yonsei.ac.kr

**Noseong Park**
Yonsei University
noseong@yonsei.ac.kr

**Jihoon Cho**
Samsung SDS
jihoon1.cho@samsung.com

## Abstract

Tabular data synthesis has received wide attention in the literature. This is because available data is often limited, incomplete, or cannot be obtained easily, and data privacy is becoming increasingly important. In this work, we present a generalized GAN framework for tabular synthesis, which combines the adversarial training of GANs and the negative log-density regularization of invertible neural networks. The proposed framework can be used for two distinctive objectives. First, we can further improve the synthesis quality, by decreasing the negative log-density of real records in the process of adversarial training. On the other hand, by increasing the negative log-density of real records, realistic fake records can be synthesized in a way that they are not too much close to real records and reduce the chance of potential information leakage. We conduct experiments with real-world datasets for classification, regression, and privacy attacks. In general, the proposed method demonstrates the best synthesis quality (in terms of task-oriented evaluation metrics, e.g., F1) when decreasing the negative log-density during the adversarial training. If increasing the negative log-density, our experimental results show that the distance between real and fake records increases, enhancing robustness against privacy attacks.

## 1 Introduction

Generative models, such as generative adversarial networks (GANs) and variational autoencoders (VAEs), have proliferated over the past several years [24, 15, 31, 2, 18, 1, 16]. GANs are one of the most successful models among generative models, and tabular data synthesis is one of the many GAN applications [7, 4, 28, 27, 21, 38].

However, tabular data synthesis is challenging due to the following two problems: i) Tabular data frequently contains sensitive information. ii) It is required to share tabular data with people, some of whom are trustworthy while others are not. Therefore, generating fake records as similar as possible to real ones, which is commonly accepted to enhance the synthesis quality, is not always preferred in tabular data synthesis, e.g., sharing with unverified people [27, 5].

In [5], it was revealed that one can effectively extract privacy-related information from a pre-trained GAN model if its log-densities are high enough. To this end, we propose a generalized framework, called *invertible tabular GAN* (IT-GAN), where we integrate the adversarial training of GANs and

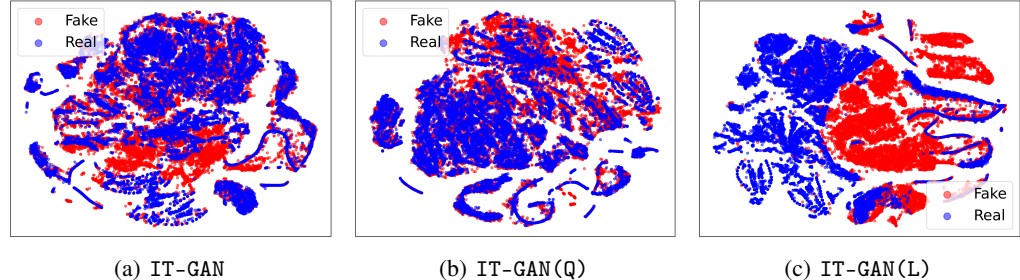

|       (a) IT-GAN       |       (b) IT-GAN(Q)       |       (c) IT-GAN(L)       |

Figure 1: Synthesis examples with IT-GAN's variations for Census, a binary classification dataset. We use t-SNE [37] to project each real/fake record onto a 2-dim space. **(a)** The synthesis only with the adversarial training shows a reasonable synthesis outcome. **(b)** The synthesis with the adversarial training and decreasing the negative log-density shows a better similarity between the distributions of the blue (real) and the red (fake) points than that of (a). **(c)** The synthesis with the adversarial training and increasing the negative log-density improves the privacy protection, i.e., it is not likely that the blue points are correctly inferred from the red points. Refer to Appendix B for additional figures.

the negative log-density training of invertible neural networks. In our framework, we can improve[1] or sacrifice the negative log-density during the adversarial training to trade off between synthesis quality and privacy protection (cf. Figure 1).

Our generator based on neural ordinary differential equations (NODEs [6]) is invertible and enables the proposed training concept. For NODEs, there exists an efficient unbiased estimation technique of their Jacobian-determinants [6, 16]. By using the unbiased Hutchinson estimator, therefore, we can efficiently estimate the negative log-density. After that, this negative log-density can be used for the following two opposite objectives: i) decreasing the negative log-density to further increase the synthesis quality (cf. IT-GAN(Q) in Figure 1), or ii) increasing the negative log-density to synthesize realistic fake records that are not too much similar to real records (cf. IT-GAN(L) in Figure 1). In particular, the second objective to make the log-density worse after little sacrificing the synthesis quality is closely related to the information leakage issue of tabular data.

However, this invertible generator has one limitation – the dimensionality of hidden layers cannot be changed. To overcome this limitation, we propose a joint architecture of an autoencoder (AE) and a GAN. The motivation behind the proposed joint architecture is twofold: i) The role of the AE is to create a hidden representation space, on which the generator and the discriminator work. The hidden representation space has the same dimensionality as that of the latent input vector of the generator. Therefore, the input and output sizes are the same in our generator, which meets the invariant dimensionality requirement of NODEs. ii) Separating the labor between the AE and the generator can improve the training process. In general, tabular data contains a large number of columns, which makes the synthesis more difficult. In our joint architecture, the generator shares its task with the AE; it does not directly synthesize fake records but fake hidden representations. The decoder (recovery network) of the AE converts them into human-readable fake records. Therefore, our final training consists of the GAN training, the AE training, and the negative log-density regularization.

We conduct experiments with 6 real-world tabular datasets and compare our method with 9 baseline methods. In many evaluation cases, our methods outperform all other baselines. Our contributions can be summarized as below:

1. We propose a general framework where we can trade-off between synthesis quality and information leakage.
2. To this end, we combine the adversarial training of GANs and the negative log-density training of invertible neural networks.
3. We conduct thorough experiments with 6 real-world tabular datasets and our methods outperform existing methods in almost all cases.

---

[1]It is well known that VAEs generate blurred samples but achieve a better log-density than GANs. Therefore, there exists a room to improve the log-density of fake samples by GANs.

## 2   Related Work

We introduce various tabular data synthesis methods and invertible neural networks. In particular, we review the invertible characteristic of NODEs.

### 2.1   Table Data Synthesis

Let $\boldsymbol{X}_{1:N}$ be a tabular data which consists of $N$ columns. Each column of $\boldsymbol{X}_{1:N}$ is either discrete or continuous numerical. Let $\boldsymbol{x}$ be a record of $\boldsymbol{X}_{1:N}$. The goal of tabular data synthesis is i) to learn a density $\hat{p}(\boldsymbol{x})$ that best approximates the data distribution $p(\boldsymbol{x})$ and ii) to generate a fake tabular data $\hat{\boldsymbol{X}}_{1:N}$. For simplicity without loss of generality, we assume $|\boldsymbol{X}_{1:N}| = |\hat{\boldsymbol{X}}_{1:N}|$, i.e., the real and fake tabular data have the same number of records.

This take can be accomplished by various approaches, e.g., VAEs, GANs, and so forth, to name a few. We introduce a couple of seminal models in this field. Tabular data synthesis, which generates a realistic synthetic table by modeling a joint probability distribution of columns in a table, encompasses many different methods depending on the types of data. For instance, Bayesian networks [3, 42] and decision trees [30] are used to generate discrete variables. A recursive modeling of tables using the Gaussian copula is used to generate continuous variables [29]. A differentially private algorithm for decomposition is used to synthesize spatial data [9, 41]. However, some constraints that these models have such as the type of distributions and computational problems have hampered high-fidelity data synthesis.

In recent years, several data generation methods based on GANs have been introduced to synthesize tabular data, which mostly handle continuous/discrete numerical records. `RGAN` [13] generates continuous time-series healthcare records while `MedGAN` [7], `CorrGAN` [28] generate discrete discrete records. `EhrGAN` [4] generates plausible labeled records using semi-supervised learning to augment limited training data. `PATE-GAN` [21] generates synthetic data without endangering the privacy of original data. `TableGAN` [27] improved tabular data synthesis using convolutional neural networks to maximize the prediction accuracy on the label column. `TGAN` [38] is one of the most recent conditional GAN-based models. It suggested a couple of important directions toward high-fidelity tabular data synthesis, e.g., a preprocessing mechanism to convert each record into a form suitable for GANs.

### 2.2   Invertible Neural Networks and Neural Ordinary Differential Equations

Invertible neural networks are typically *bijective*. Owing to this property and the change of variable theorem, we can efficiently calculate the *exact* log-density of data sample $\boldsymbol{x}$ as follows:

$$\log p_{\boldsymbol{x}}(\boldsymbol{x}) = \log p_{\boldsymbol{z}}(\boldsymbol{z}) - \log \det \left| \frac{\partial f(\boldsymbol{z})}{\partial \boldsymbol{z}} \right|, \tag{1}$$

where $f : \mathbb{R}^{\dim(\boldsymbol{z})} \to \mathbb{R}^{\dim(\boldsymbol{x})}$ is an invertible function, and $\boldsymbol{z} \sim p_{\boldsymbol{z}}(\boldsymbol{z})$. $\frac{\partial f(\boldsymbol{z})}{\partial \boldsymbol{z}}$ is the Jacobian of $f$, which is the most computationally demanding part to calculate — it has a cubic time complexity. Therefore, invertible neural networks typically restrict the Jacobian matrix definition into a form that can be efficiently calculated [31, 36, 25, 26, 10, 11]. However, FFJORD [16] recently proposed a NODE-based invertible architecture where we can use any form of Jacobian and we also rely on this technique.

In NODEs [6], let $\boldsymbol{h}(t)$ be a hidden vector at time (or layer) $t$ in a neural network. NODEs solve the following integral problem to calculate $\boldsymbol{h}(t_{i+1})$ from $\boldsymbol{h}(t_i)$ [6]:

$$\boldsymbol{h}(t_{i+1}) = \boldsymbol{h}(t_i) + \int_{t_i}^{t_{i+1}} f(\boldsymbol{h}(t), t; \boldsymbol{\theta}_f) dt, \tag{2}$$

where $f(\boldsymbol{h}(t), t; \boldsymbol{\theta}_f)$, which we call *ODE function*, is a neural network to approximate $\dot{\boldsymbol{h}} \stackrel{\text{def}}{=} \frac{d\boldsymbol{h}(t)}{dt}$. To solve the integral problem, NODEs rely on ODE solvers, e.g., the explicit Euler method, the Dormand–Prince method, and so forth [12]. $\boldsymbol{h}(t_i)$ is easily reconstructed from $\boldsymbol{h}(t_{i+1})$ with a reverse-time ODE as follows:

$$\boldsymbol{h}(t_i) = \boldsymbol{h}(t_{i+1}) + \int_{t_{i+1}}^{t_i} f(\boldsymbol{h}(t), t; \boldsymbol{\theta}_f) dt. \tag{3}$$

Two other related papers are `FlowGAN` [17] and `TimeGAN` [39] although they do not synthesize tabular data. `FlowGAN` combines the adversarial training with `NICE` [10] or `RealNVP` [11]. Grover et al. showed in the paper that likelihood-based training does not show reliable synthesis for high-dimensional space and they attempt to combine them [17]. `TimeGAN` also combines, given time-series data, the adversarial training and the supervised likelihood training of predicting a next value from past values. This supervised training is available because they deal with time-series data. In our case, we combine the adversarial training and the negative log-density regularization of NODEs, which are considered as more general than `NICE` and `RealNVP` [16].

## 3 Proposed Method

We propose a more advanced setting for tabular data synthesis than those of existing methods. Our goal is to integrate the adversarial training of GANs and the log-density training of invertible neural networks. Therefore, one can easily trade off between synthesis quality and privacy protection.

We describe our design in this section. The two key points in our design are that i) we use an invertible neural network architecture to design our generator, and ii) we integrate an AE into our framework to ii-a) enable the isometric architecture of the generator, i.e., the dimensions of hidden layers do not vary, and ii-b) distribute the workload of the generator.

### 3.1 Overall Architecture

The overall architecture is in Figure 2. Our architecture can be classified into the following four data paths: **AE-path)** The AE data path, highlighted in red, is related to an AE model to generate, given a real record $\boldsymbol{x}_{real}$, a hidden representation $\boldsymbol{h}_{real}$ and reconstruct $\hat{\boldsymbol{x}}_{real}$. **Log-density-path)** There are two different data paths related to the generator. The log-density path, highlighted in blue, is related to an invertible model to calculate the log-density of the hidden representation $\boldsymbol{h}_{real}$, i.e., $\log \hat{p}_g(\boldsymbol{h}_{real})$. Therefore, we can consider

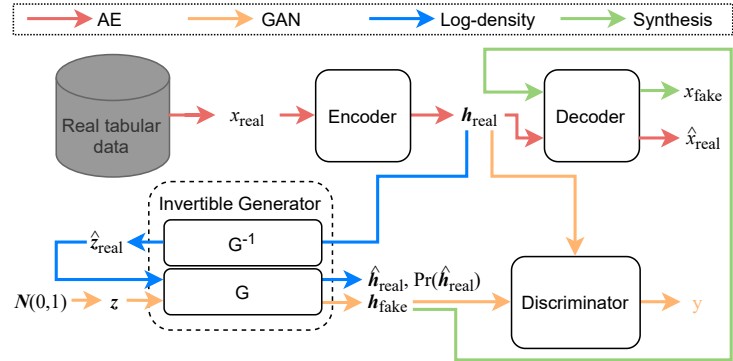

Figure 2: The overall architecture of `IT-GAN`. Each edge color means a certain type of data path.

it as the log-density of the real record $\boldsymbol{x}_{real}$ because the hidden representation is from the real record. **GAN-path)** The second data path related to the generator, highlighted in orange, is for the adversarial training. The discriminator reads $\boldsymbol{h}_{real}$ and $\boldsymbol{h}_{fake}$, which is generated by the generator from the latent vector $\boldsymbol{z}$, to distinguish them. **Synthesis-path)** The last data path, highlighted in green, is used after finishing training our model. Using the generator and the decoder, we synthesize many fake records.

### 3.2 Autoencoder

We describe our AE model in this subsection. Our AE model relies on the mode-specific normalization, which i) fits a variational mixture of Gaussians for each continuous column of $\boldsymbol{X}_{1:N}$, ii) converts each continuous element of $j$-th record $\boldsymbol{x}_{j,1:N} \in \boldsymbol{X}_{1:N}$ into a one-hot vector denoting the specific Gaussian that best matches the element and its scalar normalized value in the selected Gaussian. If a column is discrete, we simply convert each value in the column into a one-hot vector. After that we use the following encoder $\mathtt{E} : \mathcal{R}^{\dim(\boldsymbol{x})} \to \mathcal{R}^{\dim(\boldsymbol{h})}$ and the decoder (recovery network) $\mathtt{R} : \mathcal{R}^{\dim(\boldsymbol{h})} \to \mathcal{R}^{\dim(\boldsymbol{x})}$:

$$\boldsymbol{h}_{real} = \mathtt{FC1}_{n_e}(...\phi(\mathtt{FC1}_1(\boldsymbol{x}_{real}))...), \text{ for the encoder,} \tag{4}$$

$$\hat{\boldsymbol{x}}_{real} = \mathtt{FC2}_{n_r}(...\phi(\mathtt{FC2}_1(\boldsymbol{h}_{real}))...), \text{ for the decoder,} \tag{5}$$

where $\phi$ is a ReLU, $\text{FC1}_1$ is fully connected layer which takes the $\dim(\boldsymbol{x})$ size of input. Through several fully connected layers, $\text{FC1}_{n_e}$ makes the $\dim(\boldsymbol{h})$ size of output. We use $\boldsymbol{\theta}_e$ to denote the parameters of the encoder. Analogously, $\text{FC2}_1$ takes the $\dim(\boldsymbol{h})$ size of input, and after several FC layers $\text{FC2}_{n_r}$ makes the size $\dim(\boldsymbol{x})$ of output. We use $\boldsymbol{\theta}_r$ to denote the parameters of the decoder (recovery network).

### 3.3 Generator

The key point in our generator design is to adopt an invertible neural network [16] for our own purposes. Among various invertible architectures, we adopt and customize NODEs. The NODE-based invertible models have the advantage that there are no restrictions on the form of Jacobian-determinant. Other invertible models typically restrict their Jacobian-determinants to specific forms that can be easily calculated. However, our NODE-based model does not have such restrictions.

Therefore, we could study about the generator architecture without any restrictions and finally use the following architecture for our generator $\text{G} : \mathcal{R}^{\dim(\boldsymbol{z})} \to \mathcal{R}^{\dim(\boldsymbol{h})}$:

$$
\begin{aligned}
\boldsymbol{h}_{fake} &= \boldsymbol{z}(0) + \int_0^1 f(\boldsymbol{z}(t), t; \boldsymbol{\theta}_g) dt, \\
f(\boldsymbol{z}, t; \boldsymbol{\theta}_g) &= f'(\boldsymbol{z}, t; \boldsymbol{\theta}_g) - \boldsymbol{z}(t), \\
f'(\boldsymbol{z}, t; \boldsymbol{\theta}_g) &= \text{F}_K(\cdots \sigma(\text{F}_1(\boldsymbol{z}, t)) \cdots), \\
\text{F}_i(\boldsymbol{z}, t) &= (1 - \text{M}_i(\boldsymbol{z}, t))\text{FC}_{(i,1)}(\boldsymbol{z}) + \text{M}_i(\boldsymbol{z}, t)\text{FC}_{(i,2)}(\boldsymbol{z}),
\end{aligned}
\tag{6}
$$

where $\sigma$ is a non-linear activation, $\text{FC}_{(1,1)}$ and $\text{FC}_{(1,2)}$ are with $(\dim(\boldsymbol{z}), M\dim(\boldsymbol{z}))$, $\text{FC}_{(K,1)}$ and $\text{FC}_{(K,2)}$ are with $(M\dim(\boldsymbol{z}), \dim(\boldsymbol{h}))$, and $\text{FC}_{(i,1)}$ and $\text{FC}_{(i,2)}$ are with $(M\dim(\boldsymbol{z}), M\dim(\boldsymbol{z}))$ if $1 < i < K$. $\boldsymbol{z}(0) \sim \mathcal{N}(\boldsymbol{0}, \boldsymbol{1})$ is a latent vector sampled from the unit Gaussian. $\text{M}_i(\boldsymbol{z}, t) : \mathcal{R}^{\dim(\boldsymbol{z})} \to [0, 1]$ is the mapping function which defines a proportion between $\text{FC}_{(i,1)}$ and $\text{FC}_{(i,2)}$. We use either $t$ or $\texttt{sigmoid}(\text{FC}_{\text{M}_i}(\boldsymbol{z} \oplus t))$ for $\text{M}_i$, where $\oplus$ means concatenation. In our case, $\dim(\boldsymbol{z}) = \dim(\boldsymbol{h})$. In fact, this invariant dimensionality is one characteristic of many invertible neural networks. We use $\boldsymbol{\theta}_g$ to denote the parameters of the generator.

The integral problem can be solved by various ODE solvers. The log-probability $\log p(\boldsymbol{h}_{fake})$ can be calculated, by the unbiased Hutchinson estimator, as follows:

$$
\log p(\boldsymbol{h}_{fake}) = \log p(\boldsymbol{z}(0)) - \mathbb{E}_{p(\boldsymbol{\epsilon})}\Big[\int_0^1 \boldsymbol{\epsilon}^\intercal \frac{\partial f}{\partial \boldsymbol{z}(t)} \boldsymbol{\epsilon} dt\Big],
\tag{7}
$$

where $p(\boldsymbol{\epsilon})$ is a standard Gaussian or Rademacher distribution [19]. The time complexity to calculate the Hutchinson estimator is slightly larger than that of evaluating $f$ since the vector-Jacobian product $\boldsymbol{\epsilon}^\intercal \frac{\partial f}{\partial \boldsymbol{z}(t)}$ has the same cost as that of evaluating $f$ using the reverse-mode automatic differentiation.

One distinguished property of the generator is, given a real hidden vector $\boldsymbol{h}_{real}$, that we can reconstruct $\hat{\boldsymbol{h}}_{real}$ by $\text{G}(\text{G}^{-1}(\boldsymbol{h}_{real}))$ and estimate its log-probability $\hat{p}(\hat{\boldsymbol{h}}_{real})$, where $\boldsymbol{h}_{real} = \hat{\boldsymbol{h}}_{real}$ by exactly solving Eq. (3) — note that $\text{G}^{-1}$ is analytically defined from $\text{G}$ and requires no training.

### 3.4 Discriminator

The discriminator $\text{D} : \mathcal{R}^{\dim(\boldsymbol{h})} \to \mathcal{R}$ reads $\boldsymbol{h}_{real}$ and $\boldsymbol{h}_{fake}$ to classify them. We use the following architecture:

$$
y = \text{FC3}_{n_d}(...\Phi_a(\eta_b(\text{FC3}_1(\boldsymbol{h})))...),
\tag{8}
$$

where $\text{FC3}_1$ takes the size $\dim(\boldsymbol{h})$ of input, and after several FC layers $\text{FC3}_{n_d}$ makes the one dimension output. $\eta_b$ is a $\texttt{Leaky ReLU}$ with $b$ negative slope, and $\Phi_a$ is a dropout with a ratio of $a$. We use $\boldsymbol{\theta}_d$ to denote the parameters of the discriminator.

### 3.5 Training Algorithm

We describe how to train the proposed architecture. Since it consists of several modules and their loss functions, we first separately describe them and then, the final training algorithm.

**Loss Functions.** We introduce various loss functions we use to train our model. First, we use the following AE loss to train the encoder and the decoder model:

$$L_{AE} \stackrel{\text{def}}{=} L_{Reconstruct} + \frac{1}{2}\|\boldsymbol{h}_{real}\|^2 + \|\boldsymbol{h}_{fake} - \hat{\boldsymbol{h}}_{fake}\|^2, \tag{9}$$

where $L_{Reconstruct}$ is a typical reconstruction error loss. We note that we here want to learn a sparse encoder with the $L^2$ regularization term. $\boldsymbol{h}_{fake}$ is a hidden vector generated by the generator G. $\hat{\boldsymbol{h}}_{fake}$ is a reconstructed hidden vector by E(R($\boldsymbol{h}_{fake}$)), where E and R mean the encoder and the decoder, respectively. After some preliminary studies, we found that this loss definition provides robust training in many cases. In particular, it further stabilizes the integrity of the autoencoder in terms of the learned hidden representation space. To train the generator and the discriminator, we use the WGAN-GP loss. Then, we propose to use the following regularizer to control the log-density, which results in an adjustment of the real-fake distance:

$$R_{density} \stackrel{\text{def}}{=} \gamma \mathbb{E}\big[ -\log \hat{p}(\text{E}(\boldsymbol{x}))\big]_{\boldsymbol{x} \sim p_{data}}, \tag{10}$$

where E is the encoder, $\gamma$ is a coefficient to emphasize the regularization, and the log-density $\log \hat{p}(\text{E}(\boldsymbol{x}))$ can be calculated with Eq. (7) during G(G$^{-1}$(E($\boldsymbol{x}$))).

**Training Algorithm.** Algorithm 1 describes our training method. There is a training data $D_{train}$. We first train the encoder with the AE and WGAN-GP losses and the decoder with the AE loss (line 4). To learn a hidden vector space that is suitable for the overall synthesis process, we train the encoder with the WGAN-GP loss to help the discriminator better distinguish real and fake hidden vectors by learning a hidden vector in favor of the discriminator. By doing this, the AE and the GAN are integrated into a single framework. Then we train the discriminator with the WGAN-GP loss every $period_D$ (line 6), the generator with the WGAN-GP loss every $period_G$ (line 9). After that, the generator is trained one more time with the proposed density regularizer every $period_L$ (line 12). Since the discriminator and the generator rely on the hidden vector created by the AE model, we then train the encoder and the decoder every iteration. The log-density regularization is also not always used but every $period_L$ iteration because

---

**Algorithm 1:** How to train IT-GAN

**Input:** Training data $D_{train}$, Validating data $D_{val}$, Maximum iteration number $max\_iter$, The training periods $period_D, period_G, period_L$

1   Initialize $\boldsymbol{\theta}_e, \boldsymbol{\theta}_r, \boldsymbol{\theta}_g, \boldsymbol{\theta}_d$, and $k \leftarrow 0$;
2   **while** $k < max\_iter$ **do**
3      $k \leftarrow k + 1$;
4      Train $\boldsymbol{\theta}_e$ and $\boldsymbol{\theta}_r$ with $L_{AE}$ and $L_{GAN}$;
5      **if** $k \mod period_D = 0$ **then**
6         Train $\boldsymbol{\theta}_d$ with $L_{GAN}$;
7      **end**
8      **if** $k \mod period_G = 0$ **then**
9         Train $\boldsymbol{\theta}_g$ with $L_{GAN}$;
10     **end**
11     **if** $k \mod period_L = 0$ **then**
12        Train $\boldsymbol{\theta}_g$ with $R_{density}$;
13     **end**
14     Validate and update the best parameters, $\boldsymbol{\theta}_e^*, \boldsymbol{\theta}_r^*, \boldsymbol{\theta}_g^*$, and $\boldsymbol{\theta}_d^*$, with $D_{val}$;
15   **end**
16   **return** $\boldsymbol{\theta}_e^*, \boldsymbol{\theta}_r^*, \boldsymbol{\theta}_g^*$, and $\boldsymbol{\theta}_d^*$;

---

we found that a frequent log-density regularization negatively affects the entire training progress. Using the validation data $D_{val}$ and a task-oriented evaluation metric, we choose the best model. For instance, we use the F-1/MSE score of a trained generator every epoch with the validating data — an epoch consists of many iterations depending on the number of records in the training data and a mini-batch size. If a recent model is better than the temporary best model, we update it.

**On the Tractability of Training the Generator.** The ODE version of the Cauchy–Kowalevski theorem states that, given $f = \frac{d\boldsymbol{h}(t)}{dt}$, there exists a unique solution of $\boldsymbol{h}$ if $f$ is analytic (or locally Lipschitz continuous). In other words, the ODE problem is well-posed if $f$ is analytic [14]. In our case, the function $f$ in Eq. (6) uses various FC layers that are analytic and some non-linear activations that may or may not be analytic. However, the hyperbolic tangent (tanh), which is analytic, is mainly used in our experiments. This implies that there will be only one unique optimal ODE for the generator, given a latent vector $\boldsymbol{z}$. Because of i) the uniqueness of the solution and ii) our relatively simpler definitions of $f$ in comparison with other NODE applications, e.g., convolutional layer followed by a ReLU in [6], we believe that our training method can find a good solution for the generator.

Table 1: Classification in `Adult`

| Method | F1 | ROCAUC |
|---|---|---|
| Real | 0.66±0.00 | 0.88±0.00 |
| PrivBN | 0.43±0.02 | 0.84±0.01 |
| TVAE | 0.62±0.01 | 0.84±0.01 |
| TGAN | 0.63±0.01 | 0.85±0.01 |
| TableGAN | 0.46±0.03 | 0.81±0.01 |
| IT-GAN(Q) | **0.64±0.01** | **0.86±0.00** |
| IT-GAN(L) | **0.64±0.01** | 0.85±0.01 |
| IT-GAN | **0.64±0.01** | 0.86±0.01 |

Table 2: Classification in `Census`

| Method | F1 | ROCAUC |
|---|---|---|
| Real | 0.47±0.01 | 0.90±0.00 |
| PrivBN | 0.23±0.03 | 0.81±0.03 |
| TVAE | 0.44±0.01 | 0.86±0.01 |
| TGAN | 0.38±0.03 | 0.86±0.02 |
| TableGAN | 0.31±0.06 | 0.81±0.03 |
| IT-GAN(Q) | 0.45±0.01 | **0.89±0.00** |
| IT-GAN(L) | **0.46±0.01** | 0.88±0.01 |
| IT-GAN | 0.45±0.01 | 0.88±0.00 |

Table 3: Classification in `Credit`

| Method | Macro F1 | Micro F1 | ROCAUC |
|---|---|---|---|
| Real | 0.48±0.01 | 0.61±0.00 | 0.67±0.00 |
| Ind | 0.27±0.01 | 0.44±0.01 | 0.51±0.01 |
| PrivBN | 0.32±0.02 | 0.51±0.01 | **0.60±0.00** |
| TVAE | 0.39±0.00 | **0.57±0.00** | 0.58±0.00 |
| TGAN | 0.40±0.00 | 0.55±0.01 | 0.59±0.00 |
| MedGAN | 0.37±0.02 | 0.51±0.03 | 0.56±0.01 |
| IT-GAN(Q) | **0.41±0.01** | 0.54±0.01 | **0.60±0.00** |
| IT-GAN(L) | 0.40±0.01 | 0.55±0.01 | 0.60±0.01 |
| IT-GAN | 0.40±0.00 | 0.54±0.01 | 0.59±0.01 |

Table 4: Classification in `Cabs`

| Method | Macro F1 | Micro F1 | ROCAUC |
|---|---|---|---|
| Real | 0.65±0.00 | 0.68±0.00 | 0.78±0.00 |
| PrivBN | 0.64±0.00 | 0.67±0.00 | 0.77±0.00 |
| TVAE | 0.60±0.02 | 0.66±0.01 | 0.74±0.01 |
| TGAN | 0.64±0.00 | 0.67±0.00 | 0.76±0.00 |
| VeeGAN | 0.54±0.06 | 0.60±0.05 | 0.71±0.02 |
| IT-GAN(Q) | **0.66±0.00** | **0.69±0.00** | 0.79±0.01 |
| IT-GAN(L) | 0.66±0.01 | 0.68±0.01 | **0.79±0.00** |
| IT-GAN | 0.64±0.01 | 0.67±0.01 | 0.77±0.00 |

## 4 Experimental Evaluations

We introduce our experimental environments and results for tabular data synthesis. Experiments were done in the following software and hardware environments: UBUNTU 18.04, PYTHON 3.7.7, NUMPY 1.19.1, SCIPY 1.5.2, PYTORCH 1.8.1, CUDA 11.2, and NVIDIA Driver 417.22, i9 CPU, and NVIDIA RTX TITAN.

### 4.1 Experimental Environments

#### 4.1.1 Datasets

We test with various real-world tabular data, targeting binary/multi-class classification and regression. Their statistics are summarized in Appendix A. The list of datasets is as follows: `Adult` [32] consists of diverse demographic information in the U.S., extracted from the 1994 Census Survey, where we predict two classes of high (>$50K) and low (≤$50K) income. `Census` [33] is similar to `Adult` but it has different columns. `Credit` [40] is for bank loan status prediction. `Cabs` [34] is collected by an Indian cab aggregator service company for predicting the types of customers. `King` [23] contains house sale prices for King County in Seattle for the records between May 2014 and May 2015. `News` [22] has a heterogeneous set of features about articles published by Mashable in a period of two years, to predict the number of shares in social networks. `Adult` and `Census` are for binary classification, and `Credit` and `Cabs` are for multi-class classification. The others are for regression.

#### 4.1.2 Evaluation Methods

We generate fake tabular data and train multiple classification (SVM, DecisionTree, AdaBoost, and MLP) or regression (Linear Regression and MLP) algorithms. We then evaluate them with testing data and average their performance in terms of various evaluation metrics. This specific evaluation method was proposed in [38] and we follow their evaluation protocol strictly. We execute these procedures five times with five different seed numbers.

For our `IT-GAN`, we consider `IT-GAN(Q)` with a positive $\gamma$, which decreases the negative log-density to improve the synthesis **Q**uality, `IT-GAN(L)`, which sacrifices the log-density to decrease the information **L**eakage with a negative $\gamma$, and `IT-GAN`, which does not use the log-density regularization. In our result tables, `Real` means that we use real tabular data to train a classification/regression model. For other baselines, refer to Appendix C.

We do not report some baselines in our result tables to save spaces if their results are significantly worse than others. We refer to Appendix E for their full result tables.

Table 5: Regression in `King`

| Method | $R^2$ | Ex. Var. | MSE | MAE |
|---|---|---|---|---|
| Real | 0.50±0.11 | 0.61±0.02 | 0.14±0.03 | 0.30±0.03 |
| TVAE | 0.44±0.01 | 0.52±0.04 | 0.16±0.00 | 0.32±0.01 |
| TGAN | 0.43±0.01 | **0.60±0.00** | 0.16±0.00 | 0.32±0.00 |
| TableGAN | 0.41±0.02 | 0.46±0.03 | 0.17±0.01 | 0.33±0.01 |
| VeeGAN | 0.25±0.15 | 0.32±0.14 | 0.21±0.04 | 0.37±0.03 |
| IT-GAN(Q) | **0.59±0.00** | **0.60±0.00** | **0.12±0.00** | 0.28±0.00 |
| IT-GAN(L) | 0.53±0.01 | 0.56±0.01 | 0.13±0.00 | 0.29±0.00 |
| IT-GAN | 0.59±0.01 | 0.60±0.01 | **0.12±0.00** | **0.27±0.00** |

Table 6: Regression in `News` (Ex. Var. means explained variance.)

| Method | $R^2$ | Ex.Var | MSE | MAE |
|---|---|---|---|---|
| Real | 0.15±0.01 | 0.15±0.00 | 0.69± 0.00 | 0.63±0.01 |
| TVAE | -0.09±0.03 | 0.03±0.04 | 0.88±0.03 | 0.67±0.01 |
| TGAN | 0.06±0.02 | 0.07±0.01 | 0.76±0.01 | 0.66±0.02 |
| IT-GAN(Q) | **0.09±0.01** | 0.09±0.01 | **0.74±0.01** | 0.65±0.01 |
| IT-GAN(L) | 0.03±0.03 | 0.06±0.02 | 0.78±0.03 | 0.65±0.01 |
| IT-GAN | 0.09±0.02 | **0.10±0.01** | **0.74±0.01** | **0.64±0.00** |

### 4.1.3 Hyperparameters

We consider the following ranges of the hyperparameters: the numbers of layers in the encoder and the decoder, $n_e$ and $n_r$, are in {2, 3}. The number of layers $n_d$ of Eq. (8) is in {2, 3}. Dropout ratio $a$ is in {0, 0.5}, and the leaky relu slope $b$ is in {0, 0.2}. The non-linear activation $\sigma$ is `tanh`; the multiplication factor $M$ of Eq. (6) is in {1, 1.5}; the number of layers $K$ of Eq. (6) is in {3}; the coefficient $\gamma$ is in {-0.1, -0.014, -0.012, -0.01, 0, 0.01, 0.014, 0.05, 0.1}; the training periods, denoted $period_D$, $period_G$, $period_L$ in Algorithm 1, are in {1, 3, 5, 6}; the dimensionality of hidden vector $\dim(\boldsymbol{h})$ is in {32, 64, 128}; the mini-batch size is in {2000}. We use the training/validating method in Algorithm 1. For baselines, we consider the recommended set of hyperparameters in their papers or in their respected GitHub repositories. Refer to Appendix D for the best hyperparameter sets.

### 4.2 Experimental Results

In the result tables, the best (resp. second best) results are highlighted in boldface (resp. with underline). If same average, a smaller std. dev. wins. In 17 out of the 18 cases (# datasets × # task-oriented evaluation metrics), one of our methods shows the best performance in `Adult`.

**Binary Classification.** We describe the experimental results of `Adult` and `Census` in Tables 1 and 2. They are binary classification datasets. In general, many methods show reasonable evaluation scores except `Ind`, `Uniform`, and `VeeGAN`. While `TGAN` and `TVAE` show reasonable performance in terms of F1 and ROCAUC for `Adult`, our method `IT-GAN(Q)` shows the best performance overall. Among those baselines, `TGAN` shows good performance.

For `Census`, `TVAE` still works well. Whereas `TGAN` shows good performance for `Adult`, it does not show reasonable performance in `Census`. Our method outperforms them. In general, `IT-GAN(Q)` is the best in `Census`.

**Multi-class Classification.** `Credit` and `Cabs` are multi-class classification datasets, for which it is challenging to synthesize. For `Credit` in Table 3, `IT-GAN(Q)` shows the best performance in terms of Macro F1 and ROCAUC, and `TVAE` shows the best Micro F1 score. This is caused by the class imbalance problem where the minor class occupies a portion of 9%. `TVAE` doesn't create any records for it and achieves the best Micro F1 score. Therefore, `IT-GAN(Q)` is the best in `Credit`. In Figure 4 in Appendix B, `IT-GAN(L)` actively synthesizes fake records that are not overlapped with real records, which results in sub-optimal outcomes.

For `Cabs` in Table 4, `IT-GAN(Q)` shows the best scores in terms of Micro/Macro F1. Interestingly, `IT-GAN(L)` has the second best scores. From this, we can know that the sacrifice caused by increasing the negative log-density is not too much in this dataset.

**Regression.** `King` and `News` are regression datasets. Many methods show poor qualities in these datasets and tasks, and we removed them from Tables 5 and 6. In particular, all methods except `TGAN`, `IT-GAN` and its variations show negative scores for $R^2$ in `News`. Only our methods show reliable syntheses in all metrics. For `King`, our method and `TGAN` show good performance, but `IT-GAN(Q)` outperforms all baselines for almost all metrics.

**Privacy Attack.** In [5], a full black-box privacy attack method to GANs has been proposed. We implemented their method to attack our method and measure the attack success score in terms of ROCAUC. In Table 7, we report the full black-box attack success scores for our method only. Refer to Appendix G for more detailed results. In most of the cases, `IT-GAN(L)` shows the lowest attack success score. This specific `IT-GAN(L)` is the one we used to report the performance in other tables.

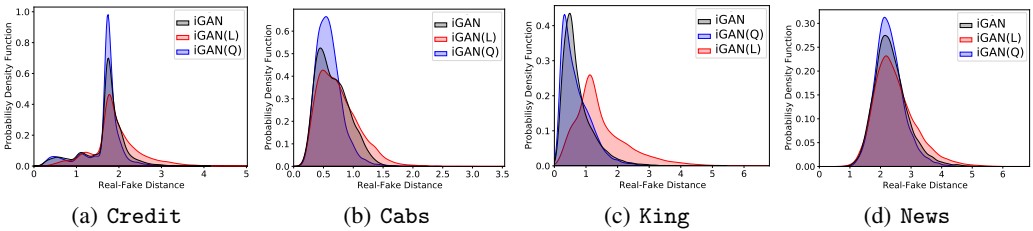

|  (a) Credit | (b) Cabs | (c) King | (d) News |

Figure 3: The p.d.f. of the real-fake distance

Table 7: ROCAUC of the full black box attack success. Lower values are more robust to attacks. If same average, lower std. dev. wins.

| Model | Adult | Census | Credit | Cabs | King | News |
|---|---|---|---|---|---|---|
| IT-GAN(Q) | 0.612±0.008 | 0.833±0.011 | 0.710±0.012 | 0.659±0.016 | 0.761±0.025 | 0.791±0.003 |
| IT-GAN(L) | **0.599±0.016** | **0.741±0.027** | **0.656±0.027** | **0.630±0.011** | **0.703±0.032** | **0.783±0.010** |
| IT-GAN | 0.618±0.003 | 0.816±0.019 | 0.688±0.058 | 0.654±0.033 | 0.742±0.003 | 0.788±0.007 |

Table 8: Sensitivity in News

| $\dim(\boldsymbol{h})$ | $R^2$ | Ex.Var | MSE | MAE |
|---|---|---|---|---|
| 32 | 0.00 | 0.01 | 0.80 | 0.67 |
| 64 | 0.03 | 0.05 | 0.78 | 0.66 |
| 128 | **0.06** | **0.07** | **0.76** | **0.65** |

Table 9: Sensitivity in News

| $\gamma$ | $R^2$ | Ex.Var | MSE | MAE |
|---|---|---|---|---|
| -0.0105 | 0.05 | 0.07 | 0.77 | 0.66 |
| -0.0100 | 0.06 | 0.07 | 0.76 | 0.65 |
| 0.0000 | 0.07 | 0.10 | 0.75 | **0.64** |
| 0.0100 | **0.10** | **0.11** | **0.73** | **0.64** |
| 0.0500 | **0.10** | 0.10 | **0.73** | 0.65 |

Table 10: Sensitivity of full black box attack w.r.t. $\gamma$ in News

| $\gamma$ | FBB ROCAUC |
|---|---|
| -0.012 | 0.762 |
| -0.011 | **0.752** |
| 0.000 | 0.784 |
| 0.050 | 0.787 |
| 0.100 | 0.792 |

Note that IT-GAN(L) shows good performance for classification and regression while having the lowest attack success score.

**Ablation Study on Negative Log-Density.** We compare IT-GAN and its variations. IT-GAN(Q) outperforms IT-GAN in Adult, Census, Credit, Cabs, and King. According to these, we can conclude that decreasing the negative log-density improves task-oriented evaluation metrics.

In Figure 3, we show the density function of the real-fake distance in various datasets whereas their mean values are shown in other experimental result tables. IT-GAN(L) effectively regularizes the distance. IT-GAN(Q) shows more similar distributions than IT-GAN. Therefore, we can know that controlling the negative log-density works as intended. The visualization in Figure 1 also proves it.

**Sensitivity Analyses.** By changing the two key hyperparameters $\dim(\boldsymbol{h})$ and $\gamma$ in our methods, we also conduct sensitivity analyses. We test IT-GAN(L) with various settings for $\dim(\boldsymbol{h})$. In general, $\dim(\boldsymbol{h}) = 128$ produces the best result as shown in Table 8. With IT-GAN in Table 9, we variate $\gamma$, and $\gamma = 0.01$ produces many good outcomes. In Table 10, $\gamma = -0.011$ is robust to the full black-box attack. We refer to Appendix F for other tables.

## 5 Conclusions

We tackled the problem of synthesizing tabular data with the adversarial training of GANs and the negative log-density regularization of invertible neural networks. Our experimental results show that the proposed methods work well in most cases and the negative log-density regularization can adjust the trade-off between the synthesis quality and the robustness to the privacy attack. However, we found that some datasets are challenging to synthesize, i.e., all generative models show lower performance than Real in some multi-class and/or imbalanced datasets, e.g., Census and Credit. In addition, the best performing method varies from one dataset/task to another, and there still exists a room to improve qualities for them.

**Societal Impacts & Limitations** Our research will foster more actively sharing and releasing tabular data. One can use our method to synthesize fake data but it is unclear how the adversary can benefit

from our research. At the same time, however, there exists a room to improve the quality of tabular data synthesis. It is still under-explored whether fake tabular data can be used for general machine learning tasks (although we showed that they can be used for classification and regression).

## Acknowledgements

Jaehoon Lee and Jihyeon Hyeong equally contributed. Noseong Park is the corresponding author. This work was supported by the Institute of Information & Communications Technology Planning & Evaluation (IITP) grant funded by the Korean government (MSIT) (No. 2020-0-01361, Artificial Intelligence Graduate School Program (Yonsei University)).

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
