

(a) IT-GAN     (b) IT-GAN(Q)     (c) IT-GAN(L)

Figure 4: Synthesis examples with IT-GAN's variations for Credit. We use t-SNE [37] to project each real/fake record onto a 2-dim space.

# A    Datasets

We summarize the statistics of our datasets as follows:

1. Adult has 22K training, 10K testing records with 6 continuous numerical, 8 categorical, and 1 discrete numerical columns.

2. Census has 200K training, 100K testing records with 7 continuous numerical, 34 categorical, and 0 discrete numerical columns.

3. Credit has 30K training records, 5K testing records with 13 continuous numerical, 4 categorical, and 0 discrete numerical columns.

4. Cabs has 40K training records, 5K testing records with 8 continuous numerical, 5 categorical, and 0 discrete numerical columns.

5. King has 16K training records, 5K testing records with 14 continuous numerical, 2 categorical, and 0 discrete numerical columns.

6. News has 32K training records, 8K testing records with 45 continuous numerical, 14 categorical, and 0 discrete numerical columns.

# B    Additional Synthesis Visualization

We introduce one more visualization with Credit in Figure 4. IT-GAN(Q) shows the best similarity between the real and fake points.

# C    Baseline Methods

We compare our method with the following baseline methods, including state-of-the-art VAEs and GANs for tabular data synthesis and our IT-GAN's three variations:

1. Ind is a heuristic method that we independently sample a value from each column's ground-truth distribution.

2. Uniform is to independently sample a value from a uniform distribution $\mathcal{U}(min, max)$ for each column, where $min$ and $max$ means the min and max value of the column.

3. CLBN [8] is a Bayesian network built by the Chow-Liu algorithm representing a joint probability distribution.

4. PrivBN [42] is a differentially private method for synthesizing tabular data using Bayesian networks.

5. MedGAN [7] is a GAN that generates discrete medical records by incorporating non-adversarial losses.

6. VeeGAN [35] is a GAN that generates tabular data with an additional reconstructor network to avoid mode collapse.

7. `TableGAN` [27] is a GAN that generates tabular data using convolutional neural networks.

8. `TVAE` [20] is a variational autoencoder (VAE) model to generate tabular data.

9. `TGAN` [38] is a GAN that generates tabular data with mixed types of variables. We use these baselines' hyperparameters recommended in their original paper and/or GitHub repositories.

## D   Best Hyperparameters for Reproducibility

We list the best hyperparameter configurations for our methods in each dataset. In all cases, we observed the best performance with mini-batch size of 2,000, and $\dim(\boldsymbol{h})$ size of 128.

1. `Adult`,
   - For `IT-GAN(Q)`, $n_e = 2, n_r = 2, n_d = 2, a = 0.5, b = 0.2, M_i(z, t) = t, M = 1, K = 3, \gamma = 0.05, period_D = 1, period_G = 1, period_L = 6$
   - For `IT-GAN(L)`, $n_e = 2, n_r = 2, n_d = 2, a = 0.5, b = 0.2, M_i(z, t) = t, M = 1, K = 3, \gamma = -0.014, period_D = 1, period_G = 1, period_L = 6$
   - For `IT-GAN`, $n_e = 2, n_r = 2, n_d = 2, a = 0.5, b = 0.2, M_i(z, t) = t, M = 1, K = 3, \gamma = 0, period_D = 1, period_G = 1, period_L = 6$

2. `Census`,
   - For `IT-GAN(Q)`, $n_e = 2, n_r = 2, n_d = 2, a = 0.5, b = 0.2, M_i(z, t) = t, M = 1, K = 3, \gamma = 0.1, period_D = 1, period_G = 1, period_L = 6$
   - For `IT-GAN(L)`, $n_e = 2, n_r = 2, n_d = 2, a = 0.5, b = 0.2, M_i(z, t) = t, M = 1, K = 3, \gamma = -0.1, period_D = 1, period_G = 1, period_L = 6$
   - For `IT-GAN`, $n_e = 2, n_r = 2, n_d = 2, a = 0.5, b = 0.2, M_i(z, t) = t, M = 1, K = 3, \gamma = 0, period_D = 1, period_G = 1, period_L = 6$

3. `Credit`,
   - For `IT-GAN(Q)`, $n_e = 2, n_r = 2, n_d = 2, a = 0, b = 0.2, M_i(z, t) = \text{sigmoid}(\text{FC}_{\text{M}_i}(\boldsymbol{z} \oplus t)), M = 1.5, K = 3, \gamma = 0.05, period_D = 5, period_G = 5, period_L = 6$
   - For `IT-GAN(L)`, $n_e = 2, n_r = 2, n_d = 2, a = 0, b = 0.2, M_i(z, t) = t, M = 1.5, K = 3, \gamma = -0.014, period_D = 1, period_G = 3, period_L = 6$
   - For `IT-GAN`, $n_e = 2, n_r = 2, n_d = 2, a = 0, b = 0.2, M_i(z, t) = t, M = 1.5, K = 3, \gamma = 0, period_D = 3, period_G = 3, period_L = 6$

4. `Cabs`,
   - For `IT-GAN(Q)`, $n_e = 2, n_r = 2, n_d = 2, a = 0, b = 0, M_i(z, t) = t, M = 1.5, K = 3, \gamma = 0.05, period_D = 3, period_G = 3, period_L = 6$
   - For `IT-GAN(L)`, $n_e = 2, n_r = 2, n_d = 2, a = 0, b = 0, M_i(z, t) = t, M = 1.5, K = 3, \gamma = -0.014, period_D = 1, period_G = 3, period_L = 6$
   - For `IT-GAN`, $n_e = 2, n_r = 2, n_d = 2, a = 0, b = 0, M_i(z, t) = t, M = 1.5, K = 3, \gamma = 0, period_D = 3, period_G = 3, period_L = 6$

5. `King`,
   - For `IT-GAN(Q)`, $n_e = 2, n_r = 2, n_d = 2, a = 0.2, b = 0.5, M_i(z, t) = t, M = 2.0, K = 5, \gamma = 0.014, period_D = 5, period_G = 1, period_L = 6$
   - For `IT-GAN(L)`, $n_e = 3, n_r = 3, n_d = 3, a = 0.2, b = 0.5, M_i(z, t) = t, M = 1.5, K = 5, \gamma = -0.011, period_D = 1, period_G = 1, period_L = 6$
   - For `IT-GAN`, $n_e = 3, n_r = 3, n_d = 2, a = 0.2, b = 0.5, M_i(z, t) = t, M = 1.5, K = 3, \gamma = 0, period_D = 5, period_G = 1, period_L = 6$

6. `News`,
   - For `IT-GAN(Q)`, $n_e = 2, n_r = 2, n_d = 2, a = 0.5, b = 0.2, M_i(z, t) = t, M = 1, K = 3, \gamma = 0.05, period_D = 1, period_G = 1, period_L = 1$
   - For `IT-GAN(L)`, $n_e = 2, n_r = 2, n_d = 2, a = 0.5, b = 0.2, M_i(z, t) = t, M = 1, K = 3, \gamma = -0.01, period_D = 1, period_G = 1, period_L = 6$
   - For `IT-GAN`, $n_e = 2, n_r = 2, n_d = 2, a = 0.5, b = 0.2, M_i(z, t) = t, M = 1, K = 3, \gamma = 0, period_D = 1, period_G = 1, period_L = 6$

# E  Full Experimental Results

In Tables 11 to 16, we list all results. We report the average of the earth mover's distance (EMD) between real and fake columns, which is to measure the dissimilarity of the two columns' cumulative distributions.

# F  Parameter Sensitivity Analyses

We report more tables about the parameter sensitivity in Tables 17 to 22.

# G  Experimental Evaluations with Privacy Attacks

We attack our method using the method in [5]. Chen et al. proposed an efficient and effective method, given a GAN model, to infer whether a record belongs to its original training data. In their research, the following full black-box attack model was actively studied because it most frequently occurs in real-world environments for releasing/sharing tabular data.

In the full black-box (FBB) attack, the attacker can access only to fake tabular data $\hat{X}_{1:N}$ generated by its victim model $G_v$. Given an unknown record $x$ in hand, the attacker approximates the probability that it is a member of the real tabular data $X_{1:N}$ used to train $G_v$, i.e., $p(x \in X_{1:N}|\hat{X}_{1:N})$. The attacker uses the nearest neighbor to $x$ among $\hat{X}_{1:N}$ as its reconstructed copy $\bar{x}$, and calculates a reconstruction error (e.g., Euclidean distance) between $x$ and $\bar{x}$. The idea behind the method is that if the record $x$ actually belongs to the real tabular data $X_{1:N}$, then it will be better reconstructed with a smaller error.

Using the FBB method, we attack IT-GAN, IT-GAN(Q), IT-GAN(L), and several sub-optimal versions of IT-GAN to compare the attack success scores for them. The reason why we include the sub-optimal versions of IT-GAN is that one can easily decrease the attack success score by selecting sub-optimal models (and thereby, potentially sacrificing the synthesis quality). We show that IT-GAN(L) is a more principled way to trade off between the two factors. Also, since the attacker is more likely to target high-quality models, we selectively attack TGAN and TVAE, which show comparably high synthesis quality in some cases.

To evaluate the attack success scores, we create a balanced evaluation set of the randomly-sampled training data and the original testing data from Adult, Census, and so on. If an attack correctly classifies random training samples as the original training data and testing samples as not belonging to it, it turns out to be successful. As this attack is a binary classification, we use the ROCAUC score as its evaluation metric. We train, generate and attack each generative model for five times with different seed numbers and report the average and std. dev. of their attack success scores.

As shown in Table 23, IT-GAN(L) shows the lowest attack success scores overall. For Cabs and Credit, the attack success scores of TVAE are the lowest among all methods, which means it is the most robust to the attack among all the methods for those two datasets. However, it marks relatively poor synthesis quality for them as reported in Tables 3 and 4. IT-GAN(L) shows attack success scores close to those of TVAE while it shows better machine learning tasks scores, e.g., a Macro F1 score of 0.60 by TVAE vs. 0.66 by IT-GAN(L) in Cabs. All in all, TVAE shows its robustness in the two datasets for its relatively low synthesis quality but in general, IT-GAN(L) achieves the best trade-off between synthesis quality and privacy protection.

We also compare with various sub-optimal versions of IT-GAN. While training IT-GAN, we could obtain several sub-optimal versions around the epoch where we obtain IT-GAN and call them as IT-GAN(S$k$) with $k \in \{1, 2, 3\}$. They all show sub-optimal accuracy in comparison with IT-GAN. One easy way to protect sensitive information in the original data is to use these sub-optimal versions. These IT-GAN(S$k$) sub-optimal models have F1 (resp. MSE) scores worse (resp. larger) than those of IT-GAN in Tables 1 to 6 by $0.01 \times k$ — if multiple sub-optimal models have the same score, we choose the sub-optimal model with the largest distance to increase the robustness to the attack. However, they typically fail to obtain a good trade-off between synthesis quality and information protection. Among those three sub-optimal versions, for instance, IT-GAN(S1) shows the best synthesis quality. However, IT-GAN(L) still outperforms it in 5 out of the 6 datasets in terms of the task-oriented evaluation metrics in Tables 1 to 6. Therefore, we can say that IT-GAN(L) is a more

Table 11: Classification in `Adult`

| Method | Acc. | F1 | ROCAUC | EMD |
|--------|------|----|--------|-----|
| Real | 0.83±0.00 | 0.66±0.00 | 0.88±0.00 | 0.00±0.00 |
| CLBN | 0.77±0.00 | 0.31±0.03 | 0.78±0.01 | 2.05e+03±10.28 |
| Ind | 0.65±0.03 | 0.17±0.04 | 0.51±0.08 | 27.88±2.9 |
| PrivBN | 0.79±0.01 | 0.43±0.02 | 0.84±0.01 | 3.15e+02±14.82 |
| Uniform | 0.49±0.10 | 0.26±0.09 | 0.50± 0.07 | 6.42e+03±19.13 |
| TVAE | 0.81±0.00 | 0.62±0.01 | 0.84±0.01 | 6.28e+02±1.08e+02 |
| TGAN | 0.81±0.00 | 0.63±0.01 | 0.85±0.01 | 2.22e+02±1.71e+02 |
| TableGAN | 0.80±0.01 | 0.46±0.03 | 0.81±0.01 | 1.07e+02±13.52 |
| VeeGAN | 0.69±0.04 | 0.47±0.04 | 0.73±0.04 | 1.78e+03±1.57e+02 |
| MedGAN | 0.71±0.05 | 0.48±0.05 | 0.74±0.02 | 7.06e+03±1.53e+02 |
| IT-GAN(Q) | 0.81±0.00 | 0.64±0.01 | 0.86±0.00 | 87.32±42.34 |
| IT-GAN(L) | 0.80±0.01 | 0.64±0.01 | 0.85±0.01 | 3.21e+02±72.21 |
| IT-GAN | 0.81±0.00 | 0.64±0.01 | 0.86±0.01 | 72.49±6.36 |

Table 12: Classification in `Census`

| Method | Acc. | F1 | ROCAUC | EMD |
|--------|------|----|--------|-----|
| Real | 0.91±0.00 | 0.47±0.01 | 0.90± 0.00 | 0.00±0.00 |
| CLBN | 0.90± 0.00 | 0.29±0.01 | 0.75±0.01 | 15.07±0.03 |
| Ind | 0.71±0.05 | 0.06±0.01 | 0.45±0.02 | 0.13±0.01 |
| PrivBN | 0.91±0.01 | 0.23±0.03 | 0.81±0.03 | 5.64±1.78 |
| Uniform | 0.49±0.22 | 0.11±0.01 | 0.48±0.05 | 2.08e+02±0.12 |
| TVAE | 0.93±0.00 | 0.44±0.01 | 0.86±0.01 | 1.09±0.15 |
| TGAN | 0.91±0.01 | 0.38±0.03 | 0.86±0.02 | 0.98±0.08 |
| TableGAN | 0.93±0.01 | 0.31±0.06 | 0.81±0.03 | 0.81±0.22 |
| VeeGAN | 0.85±0.07 | 0.18±0.08 | 0.67±0.08 | 1.57±0.03 |
| MedGAN | 0.78±0.13 | 0.15±0.06 | 0.64±0.09 | 2.38e+02±46.19 |
| IT-GAN(Q) | 0.91±0.00 | 0.45±0.01 | 0.89±0.00 | 1.02±0.09 |
| IT-GAN(L) | 0.91±0.01 | 0.46±0.01 | 0.88±0.01 | 3.77±1.25 |
| IT-GAN | 0.91±0.00 | 0.45±0.01 | 0.88±0.00 | 1.29±0.31 |

principled method than other methods in terms of the trade-off. `IT-GAN(L)` is able to synthesize high-quality fake records while successfully regularizing the log-density of real records.

In the following two tables (Tables 24 and 25), we show the sensitivity analyses related to the attack. By increasing or decreasing $\gamma$, our model's robustness can be properly adjusted as expected. This property is our main research goal in this paper. Our framework, which combines the adversarial training and the negative log-density regularization, provides a principled way for synthesizing tabular data.

Table 13: Classification in `Credit`

| Method | Acc. | Macro F1 | Micro F1 | ROCAUC | EMD |
|---|---|---|---|---|---|
| Real | 0.61±0.00 | 0.48±0.01 | 0.61±0.00 | 0.67±0.00 | 0.00±0.00 |
| CLBN | 0.48±0.01 | 0.34±0.02 | 0.48±0.01 | 0.56±0.01 | 1.85e+06±4.40e+03 |
| Ind | 0.44±0.01 | 0.27±0.01 | 0.44±0.01 | 0.51±0.01 | 1.13e+04±5.51e+03 |
| PrivBN | 0.51±0.01 | 0.32±0.02 | 0.51±0.01 | 0.60±0.00 | 3.93e+05±3.85e+04 |
| Uniform | 0.34±0.07 | 0.24±0.04 | 0.34±0.07 | 0.50±0.02 | 1.89e+07±4.20e+04 |
| TVAE | 0.57±0.00 | 0.39±0.00 | 0.57±0.00 | 0.58±0.00 | 4.65e+05±1.01e+05 |
| TGAN | 0.55±0.01 | 0.40±0.00 | 0.55±0.01 | 0.59±0.00 | 2.36e+05±1.95e+05 |
| TableGAN | 0.47±0.02 | 0.35±0.01 | 0.47±0.02 | 0.54±0.01 | 2.08e+05±2.30e+04 |
| VeeGAN | 0.48±0.03 | 0.36±0.02 | 0.48±0.03 | 0.54±0.02 | 1.85e+06±4.31e+05 |
| MedGAN | 0.51±0.03 | 0.37±0.02 | 0.51±0.03 | 0.56±0.01 | 1.95e+07±2.22e+06 |
| IT-GAN(Q) | 0.54±0.01 | 0.41±0.01 | 0.54±0.01 | 0.60±0.00 | 3.62e+05±2.15e+05 |
| IT-GAN(L) | 0.55±0.01 | 0.40±0.01 | 0.55±0.01 | 0.60±0.01 | 5.29e+05±3.00e+05 |
| IT-GAN | 0.54±0.01 | 0.40±0.00 | 0.54±0.01 | 0.59±0.01 | 2.38e+05±1.12e+05 |

Table 14: Classification in `Cabs`

| Method | Acc. | Macro F1 | Micro F1 | ROCAUC | EMD |
|---|---|---|---|---|---|
| Real | 0.68±0.00 | 0.65±0.00 | 0.68±0.00 | 0.78±0.00 | 0.00±0.00 |
| CLBN | 0.66±0.00 | 0.63±0.00 | 0.66±0.00 | 0.76±0.00 | 6.72±0.0 |
| Ind | 0.39±0.01 | 0.31±0.01 | 0.39±0.01 | 0.50± 0.01 | 0.05±0.0 |
| PrivBN | 0.67±0.00 | 0.64±0.00 | 0.67±0.00 | 0.77±0.00 | 0.25±0.0 |
| Uniform | 0.31±0.02 | 0.28±0.02 | 0.31±0.02 | 0.47±0.02 | 3.64±0.02 |
| TVAE | 0.66±0.01 | 0.60± 0.02 | 0.66±0.01 | 0.74±0.01 | 0.66±0.13 |
| TGAN | 0.67±0.00 | 0.64±0.00 | 0.67±0.00 | 0.76±0.00 | 0.38±0.11 |
| TableGAN | 0.41±0.02 | 0.35±0.03 | 0.41±0.02 | 0.52±0.03 | 0.21±0.06 |
| VeeGAN | 0.60± 0.05 | 0.54±0.06 | 0.60± 0.05 | 0.71±0.02 | 2.02±0.64 |
| MedGAN | 0.55±0.03 | 0.48±0.03 | 0.55±0.03 | 0.65±0.02 | 4.56±0.17 |
| IT-GAN(Q) | 0.69±0.00 | 0.66±0.00 | 0.69±0.00 | 0.79±0.01 | 0.33±0.10 |
| IT-GAN(L) | 0.68±0.01 | 0.66±0.01 | 0.68±0.01 | 0.79±0.00 | 0.63±0.11 |
| IT-GAN | 0.67±0.01 | 0.64±0.01 | 0.67±0.01 | 0.77±0.00 | 0.36±0.09 |

Table 15: Regression in `King`

| Method | $R^2$ | Ex.Var | MSE | MAE | EMD |
|---|---|---|---|---|---|
| Real | 0.50±0.11 | 0.61±0.02 | 0.14±0.03 | 0.30±0.03 | 0.00±0.00 |
| CLBN | -0.05±0.29 | 0.35±0.06 | 0.57±0.40 | 0.56±0.21 | 3.53e+04±1.60e+02 |
| Ind | -0.09±0.26 | 0.13±0.04 | 0.34±0.14 | 0.46±0.10 | 5.35e+02±51.97 |
| PrivBN | -0.01±0.20 | 0.51±0.02 | 0.31±0.09 | 0.44±0.07 | 1.30e+04±1.23e+03 |
| Uniform | -0.98±0.05 | 0.05±0.05 | 3.84±2.13 | 1.76±0.57 | 2.55e+05±1.23e+03 |
| TVAE | 0.44±0.01 | 0.52±0.04 | 0.16±0.00 | 0.32±0.01 | 2.65e+03±3.97e+02 |
| TGAN | 0.43±0.01 | 0.60± 0.00 | 0.16±0.00 | 0.32±0.00 | 5.67e+03±2.20e+03 |
| TableGAN | 0.41±0.02 | 0.46±0.03 | 0.17±0.01 | 0.33±0.01 | 3.62e+03±3.51e+02 |
| VeeGAN | 0.25±0.15 | 0.32±0.14 | 0.21±0.04 | 0.37±0.03 | 6.83e+04±5.32e+04 |
| MedGAN | -0.39±0.02 | 0.01±0.13 | 0.97±0.19 | 0.78±0.08 | 2.90e+05±3.09e+04 |
| IT-GAN(Q) | 0.59±0.00 | 0.60±0.00 | 0.12±0.00 | 0.28±0.00 | 2.04e+03±2.66e+02 |
| IT-GAN(L) | 0.53±0.01 | 0.56±0.02 | 0.13±0.00 | 0.29±0.00 | 1.63e+04±6.17e+03 |
| IT-GAN | 0.59±0.01 | 0.60±0.01 | 0.12±0.00 | 0.27±0.00 | 4.57e+03±1.27e+03 |

Table 16: Regression in `News`

| Method | $R^2$ | Ex.Var | MSE | MAE | EMD |
|---|---|---|---|---|---|
| Real | 0.15±0.01 | 0.15±0.00 | 0.69± 0.00 | 0.63±0.01 | 0.00±0.00 |
| CLBN | -1.00±0.00 | -1.08±0.56 | 5.75±0.95 | 1.99±0.18 | 1.60e+04±23.41 |
| Ind | -0.06±0.05 | -0.03±0.03 | 0.85±0.04 | 0.70± 0.01 | 1.07e+02±12.39 |
| PrivBN | -0.58±0.12 | -1.04±0.67 | 1.93±0.61 | 1.04±0.10 | 1.04e+03±18.63 |
| Uniform | -0.86±0.17 | -1.02±1.43 | 4.73±1.41 | 1.79±0.21 | 3.98e+04±28.96 |
| TVAE | -0.09±0.03 | 0.03±0.04 | 0.88±0.03 | 0.67±0.01 | 8.74e+02±1.88e+02 |
| TGAN | 0.06±0.02 | 0.07±0.01 | 0.76±0.01 | 0.66±0.02 | 4.99e+02±94.4 |
| TableGAN | -0.86±0.09 | -0.80±0.30 | 1.61±0.16 | 1.01±0.04 | 1.21e+03±3.84e+02 |
| VeeGAN | -0.91±0.15 | -1.36e+07±2.49e+07 | 1.49e+07±2.83e+07 | 9.54e+02±1.35e+03 | 4.63e+03±8.00e+02 |
| MedGAN | -0.73±0.08 | -0.19±0.08 | 3.99±0.06 | 1.64±0.03 | 3.93e+04±4.06e+03 |
| IT-GAN(Q) | 0.09±0.01 | 0.09±0.01 | 0.74±0.01 | 0.65±0.01 | 2.24e+02±62.01 |
| IT-GAN(L) | 0.03±0.03 | 0.06±0.02 | 0.78±0.03 | 0.65±0.01 | 1.02e+03±98.7 |
| IT-GAN | 0.09±0.02 | 0.10±0.01 | 0.74±0.01 | 0.64±0.00 | 4.86e+02±1.21e+02 |

Table 17: Parameter Sensitivity in `Adult`

| $\dim(\boldsymbol{h})$ | Acc. | F1 | ROCAUC | EMD |
|---|---|---|---|---|
| 32 | 0.81 | 0.64 | 0.85 | 108.71 |
| 64 | 0.81 | 0.65 | 0.86 | 67.07 |
| 128 | 0.80 | 0.65 | 0.86 | 231.45 |

Table 18: Parameter Sensitivity in `Census`

| $\dim(\boldsymbol{h})$ | Acc. | F1 | ROCAUC | EMD |
|---|---|---|---|---|
| 32 | 0.91 | 0.42 | 0.87 | 1.34 |
| 64 | 0.90 | 0.46 | 0.89 | 1.17 |
| 128 | 0.91 | 0.47 | 0.89 | 3.34 |

Table 19: Parameter Sensitivity in `News`

| $\dim(\boldsymbol{h})$ | $R^2$ | Ex.Var | MSE | MAE | EMD |
|---|---|---|---|---|---|
| 32 | 0.00 | 0.01 | 0.80 | 0.67 | 1270.44 |
| 64 | 0.03 | 0.05 | 0.78 | 0.66 | 302.38 |
| 128 | 0.06 | 0.07 | 0.76 | 0.65 | 972.37 |

Table 20: Parameter Sensitivity in `Adult`

| $\gamma$ | Acc. | F1 | ROCAUC | EMD |
|---|---|---|---|---|
| -0.018 | 0.79 | 0.64 | 0.85 | 377.11 |
| -0.014 | 0.80 | 0.65 | 0.86 | 231.45 |
| -0.010 | 0.80 | 0.64 | 0.86 | 169.16 |
| 0.000 | 0.81 | 0.64 | 0.86 | 76.13 |
| 0.010 | 0.81 | 0.65 | 0.86 | 92.04 |
| 0.050 | 0.81 | 0.65 | 0.86 | 83.74 |
| 0.100 | 0.81 | 0.64 | 0.86 | 167.99 |

Table 21: Parameter Sensitivity in `Census`

| $\gamma$ | Acc. | F1 | ROCAUC | EMD |
|---|---|---|---|---|
| -0.20 | 0.90 | 0.43 | 0.87 | 0.89 |
| -0.10 | 0.91 | 0.47 | 0.89 | 3.34 |
| -0.05 | 0.91 | 0.45 | 0.88 | 1.15 |
| 0.00 | 0.91 | 0.46 | 0.89 | 1.73 |
| 0.05 | 0.90 | 0.44 | 0.88 | 0.83 |
| 0.10 | 0.90 | 0.44 | 0.89 | 0.84 |
| 0.50 | 0.90 | 0.46 | 0.89 | 2.91 |

Table 22: Parameter Sensitivity in `News`

| $\gamma$ | $R^2$ | Ex.Var | MSE | MAE | EMD |
|---|---|---|---|---|---|
| -0.0120 | -0.06 | 0.03 | 0.85 | 0.67 | 1252.72 |
| -0.0110 | 0.02 | 0.07 | 0.79 | 0.65 | 1192.79 |
| -0.0105 | 0.05 | 0.07 | 0.77 | 0.66 | 1009.46 |
| -0.0100 | 0.06 | 0.07 | 0.76 | 0.65 | 972.37 |
| 0.000 | 0.07 | 0.10 | 0.75 | 0.64 | 620.97 |
| 0.0100 | 0.10 | 0.11 | 0.73 | 0.64 | 473.49 |
| 0.0500 | 0.10 | 0.10 | 0.73 | 0.65 | 320.15 |
| 0.1000 | 0.10 | 0.10 | 0.73 | 0.64 | 477.28 |

Table 23: Full black box attack success rates. Lower values are more robust to attacks. If same average, lower std. dev. wins. `IT-GAN(Sk)` is a sub-optimal `IT-GAN` model, which has an F1 score of $\text{F1}(\text{IT-GAN}) - 0.01 \times k$ or an MSE of $\text{MSE}(\text{IT-GAN}) + 0.01 \times k$. We obtained these sub-optimal models (checkpoints) during training `IT-GAN`.

| Model | Adult | Census | Credit | Cabs | King | News |
|-------|-------|--------|--------|------|------|------|
| TVAE | 0.721±0.033 | 0.902±0.016 | **0.646±0.016** | **0.620±0.040** | 0.760±0.014 | 0.802±0.005 |
| TGAN | 0.647±0.012 | 0.848±0.014 | 0.684±0.023 | 0.688±0.011 | 0.781±0.007 | 0.799±0.004 |
| IT-GAN(Q) | 0.612±0.008 | 0.833±0.011 | 0.710±0.012 | 0.659±0.016 | 0.761±0.025 | 0.791±0.003 |
| IT-GAN(L) | **0.599±0.016** | **0.741±0.027** | 0.656±0.027 | 0.630±0.011 | **0.703±0.032** | **0.783±0.010** |
| IT-GAN | 0.618±0.003 | 0.816±0.019 | 0.688±0.058 | 0.654±0.033 | 0.742±0.003 | 0.788±0.007 |
| IT-GAN(S1) | 0.616±0.010 | 0.820±0.018 | 0.663±0.037 | 0.647±0.018 | 0.726±0.023 | 0.785±0.007 |
| IT-GAN(S2) | 0.614±0.006 | 0.829±0.023 | 0.664±0.039 | 0.643±0.033 | 0.711±0.025 | 0.784±0.008 |
| IT-GAN(S3) | 0.616±0.007 | 0.831±0.016 | 0.664±0.033 | 0.648±0.033 | 0.727±0.026 | 0.785±0.011 |

Table 24: Sensitivity of full black box attack w.r.t. $\gamma$ in `Adult`

| $\gamma$ | FBB ROCAUC |
|----------|------------|
| -0.018 | **0.583** |
| -0.014 | 0.605 |
| -0.01 | 0.606 |
| 0.0 | 0.623 |
| 0.01 | 0.614 |
| 0.05 | 0.621 |
| 0.1 | 0.610 |

Table 25: Sensitivity of full black box attack w.r.t. $\gamma$ in `News`

| $\gamma$ | FBB ROCAUC |
|----------|------------|
| -0.012 | 0.762 |
| -0.011 | **0.752** |
| -0.0105 | 0.783 |
| -0.01 | 0.774 |
| 0.0 | 0.784 |
| 0.01 | 0.781 |
| 0.05 | 0.787 |
| 0.1 | 0.792 |