# OpenReview forum: "Invertible Tabular GANs: Killing Two Birds with One Stone for Tabular Data Synthesis"
_NeurIPS.cc/2021/Conference — NeurIPS 2021 Poster_

### Official Review · Reviewer_1ZTc · 2021-07-14

**Rating:** 7
**Confidence:** 4

**Summary:**

This paper proposes a new GAN framework for tabular data synthesis, which combines the adversarial training of GAN and negative-log density regularization of invertible networks. The negative-log density regularization enables the trade-off between synthetic quality or privacy protection. Specifically, the G is designed via the motivation of NODE to estimate the invertible latent vector from the hidden representation. The AE is trained to encode the hidden representation of real samples and GAN is trained via the hidden representation instead of raw data.

**Limitations And Societal Impact:**

The authors adequately addressed the limitations and potential negative societal impact of their work.

**Main Review:**

Pros

The paper is nicely written and easy to read. The proposed model is technically new to me and achieves state-of-the-art results on benchmark datasets.

Cons

However, the paper discusses the overall performance of the proposed model with the best hyper-parameters. It would be more useful to readers to have an ablation study on them.

Moreover, the proposed model is a bit different from the standard GAN model, thus one needs to understand how much each proposed component contributes to the performance.

Minor mistake in Line 279: periodD, periodG, periodL in Algorithm 1, are in {1, 3, 5, 6}.


**Time Spent Reviewing:**

Three hours

---

> ### Author Response · Authors · 2021-08-07
> **Thanks for your review comments.**
>
> First of all, we appreciate your efforts in reading our paper and letting us know your comments. We answer your questions as follows:
>
> - IT-GAN (Q), IT-GAN, and IT-GAN (L) can show the ablation study on the likelihood regularization, which is our main ablation study direction. We will also add one more loss definition suggested by the reviewer vTau, i.e., a convex weighted sum of the two real and fake discriminator losses with various weight combinations. We could achieve sub-optimal outcomes for it, e.g., an F-1 score of 0.62 for Adult vs. 0.64 with our method. For other datasets, it also shows sub-optimal outcomes. We will add these experimental results to our paper.
>
> - We will better highlight the role of each network in our model. We agree that our model design is a little more complicated than other standard GANs. Invertible neural networks require that the output size is the same as the input size. Our invertible generator’s latent vector input size should be the same as the output hidden vector size. The main purpose of adding the autoencoder is to match the input and the output sizes of the invertible generator.
>
> - We will correct Line 279 following your suggestion. Thanks for letting us know.

---

### Official Review · Reviewer_ydPV · 2021-07-19

**Rating:** 7
**Confidence:** 4

**Summary:**

This paper proposes the invertible tabular GAN framework which contains an auto encoder to encode the data into a vector representation, and an invertible generator to generate such representation. The invertibility of the generator enables the computation of likelihood for real data, and the likelihood can be used as a regularization to the network.


**Limitations And Societal Impact:**

I don’t think this work can lead to any negative social impact.


**Main Review:**

Originality:
- The idea of introducing invertible neural networks into synthetic data generation is a novel and interesting idea.
- However, the IT-GAN model only learns the pdf for the hidden vector generated by the auto-encoder, rather than the tabular data itself. Thus does not solve the fundamental challenge in learning the probability distribution of tabular data.

Quality:
- The performance of the proposed model has been carefully evaluated on multiple real-world datasets and compared with strong baselines. - The results show that the proposed method can outperform existing methods, and on some metrics with a large margin.
- The experiments are well-designed, including the privacy attack experiment and the distance distribution table, which can clearly show the effect of decreasing likelihood.
Clarity:
- This paper is clearly written and well-organized. Experimental settings and results are listed and code is available.

Significance:
- Although the proposed method is somewhat incremental, bringing invertible neural networks to model the probability distribution and generate synthetic data at the same time is a non-trivial idea. Such work is inspiring for future research and beneficial for the field.

Minors:
- Footnote 1: It is well known that VAEs generate blurred samples but achieve a better log-density than GANs. Such observation is made on images, while tabular data may be different.


**Time Spent Reviewing:**

3

---

> ### Author Response · Authors · 2021-08-07
> **Thanks for your review comments.**
>
> First of all, we appreciate your efforts in reading our paper and letting us know your comments. We will modify the footnote following your suggestion. We will clarify that the statement is especially for images.

---

> ### Public Comment · ~Dionysis_Manousakas1 · 2022-08-29
> **On code availability**
>
> ydPV mentions that the code of the paper is available. I went through the published version but couldn't find a related link. Is this code (still) public?

---

> > ### Public Comment · Authors · 2022-08-29
> > **Code availability**
> >
> > Samsung supported this work and we are now discussing with them to release our source codes. Could you please wait until we finalize our discussion?
> >
> > Authors

---

> > > ### Public Comment · ~Dionysis_Manousakas1 · 2022-08-29
> > > **On code availability**
> > >
> > > Sure, thanks

---

> > > > ### Public Comment · Authors · 2023-01-20
> > > > **Code released**
> > > >
> > > > Samsung reviewed our codes and we just released our codes. Plz check the public comment above. Note that our codes follows the Samsung SDS source code license.

---

### Official Review · Reviewer_vTau · 2021-07-31

**Rating:** 7
**Confidence:** 4

**Summary:**

The paper proposes a new architecture for tabular data synthesis. In particular, the proposed architecture utilizes a combination of autoencoders (AEs), generative adversarial networks (GANs), and invertible models. The architecture can also consider aspects of privacy via a hyperparameter which controls the likelihood of the generator producing data from the training set.

Main Contributions:
  - The proposed architecture, utilizes an AE to first change inputs into a latent dimension for the GAN to then process.
  - The invertible model is utilized in the generator of the GAN -> This ensures that the latent representation of the AE agrees with the invertible generator (such that there exists an input to the generator which will produce the true latent representation of an AE input)
  - Furthermore, the random generator inputs induce a pdf -> which gives a pdf on the (reconstructed) AE latent space generated by the generator network (via invertibility). This can thus be used as a regularization condition to penalise/reward the likelihood of training data's latent representations generated by the generator network.
  - The following is experimentally tested using a number of datasets and against a number of baselines, with better reported performance.
  - A privacy experiment is done. Here, the closeness of generated data to training data is evaluated.

**Ethical Concerns:**

No ethical concerns.

**Limitations And Societal Impact:**

The authors have adequately addressed the limitations and potential negative societal impact of their work.

**Main Review:**

Overall the paper, provides a novel architecture which combines a number of different ideas for synthesising tabular data. The paper is well written and clear. The ideas composition of previous architectures is also interesting. The only notable flaw is that the angle in which privacy is examined is only examined experimentally with no theoretic analysis. However, the large coverage of experiments helps provide evidence for this approach.

Originality & Significance: Although the individual components of the architecture are not novel, their composition appears to be. One fact which could hurt novelty is if the composition of AE and GAN has been used before. If so, this should be highlighted in prior work. The use of the invertible generator is also very interesting.

Quality & Clarity: The paper seems to be technically sound with a nice and extensive experimental section. The only issue is the lack of runtime and architecture size comparisons. The paper is also well written. Although the heavy notation in the specification of the architecture hurts clarity a bit.

Strengths:
  - The combination of AE, GAN, and invertible generator is neat.
  - The log-density likelihood regularizer/loss function is interesting. It seems that this can also be used to make different partitions of real world data more likely (i.e., create a specialized generator by rewarding/penalizing different subsets of the training data).

Weaknesses/Suggestions/Questions:
  - The notation of the loss function in (9) suggest that the regularity/sparsity terms only depend on the fake data. If so, why not have this also depend on the training data/real records?
  - What is the runtime compared to baselines? How does your proposed architecture compare in the number of parameters to baselines?
  - Is there a way to relate the log-density regularizer and the fake-real distance other definitions of privacy? (differential privacy?)
  - There seems to be a missing point of comparison when the log-density regularizer is not used. What happens when you don't use an invertible generator?
  - Furthermore, without the invertible generator can the log-density regularizer be substituted by just considering a convex reweighting of the discriminator error for real and fake error? (i.e., for $ \gamma > 0 $, weight the error of discriminator for realdata being lower than that of an error for fake data)

**Time Spent Reviewing:**

3

---

> ### Author Response · Authors · 2021-08-07
> **Thanks for your review comments.**
>
> First of all, we appreciate your efforts in reading our paper and letting us know your comments. We answer your questions as follows:
>
> -	In Eq. (9), we found that it is a typo. In fact, we have the sparsity regularization on real records. We wrote it as h_{fake} by a mistake. We will correct this.
> -	TGAN’s parameter size is similar to ours in many cases. However, TVAE, VeeGAN, and MedGAN have approximately 60% lesser parameters. In particular, TableGAN's size is the smallest, i.e., 90% lesser parameters in comparison with ours.
> -	It takes a second per mini-batch iteration (the mini-batch size is 2000) for our method. TGAN’s training time is 5-10 times faster than ours (because we need to solve ODEs for our method, which typically requires multiple steps) and TableGAN is 5-10 times faster than TGAN. In general, neural ordinary differential equations require longer forward and backward time than conventional neural networks.
> -	Chen et al. introduces the relationship between likelihood and membership inference attack in [5], which is one motivation of our work. However, it is unclear how we can connect it to the differential privacy for now since the differential privacy's privacy notion is more complicated.
> - Among IT-GAN (Q), IT-GAN, and IT-GAN (L), IT-GAN does not use the log-density regularization. So you can easily check the efficacy of the regularization in our result tables.
> - We constructed the loss following your suggestion, i.e., a convex weighted sum of the two real and fake discriminator losses with various weight combinations. We could achieve sub-optimal outcomes for it, e.g., an F-1 score of 0.62 for Adult vs. 0.64 with our method. For other datasets, it also shows sub-optimal outcomes.

---

> > ### Comment · Reviewer_vTau · 2021-08-25
> > **Thanks for the response + update**
> >
> > Thanks you for your response.
> > I am happy with your reply. Thank you for pointing out differences between your approaches. I somehow missed the specification of IT-GAN.
> >
> > I'll be updating my score accordingly.

---

### Official Review · Reviewer_uc8f · 2021-08-03

**Rating:** 7
**Confidence:** 4

**Summary:**

The authors present a generalised GAN framework for tabular synthesis wherein, with a view towards privacy, the Negative Log Likelihood (NLL) of the synthesised dataset can be tuned. The authors evaluate this technique of likelihood tuning with regards to tradeoffs in downstream tasks and privacy attacks. To evaluate the NLL the authors are required to use an invertible network, which imposes dimensionality constraints. To overcome these constraints the authors first push the data to a lower-dimensional, latent space using an auto encoder, on which the GAN is trained.

**Limitations And Societal Impact:**

The authors address some limitations of their approach with regards to developing synthetic datasets realistic enough to train models with the same accuracy as the original datasets. But these limitations are not unique to this work, and are likely a larger problem beyond the scope of this paper.

One question I had reading the paper was the necessity of the VAE to develop the low dimensional encoding on which the GAN is trained. Do you have any experiments showing the benefit of this added complexity? Is it feasible to compute the Jacobean determinant in the original dimension space?

**Main Review:**

While the problem and method of anonymising tabular data using a GAN appears not to be new, the proposed technique of using the NLL to guard against privacy techniques seems original. The account of the existing literature is thorough and the contribution of the authors is well defined.

The authors give cursory theoretical considerations to several aspects of their approach, in particular the time complexity for the Jacobean determinant calculations and suitably of the algorithms to solve the associated ODE problem. The experimental results are well presented, and the authors' approach appears to significantly outperform other techniques with respect to synthesising tabular data and defending against privacy attacks.

The work is very clearly written and concisely.



**Time Spent Reviewing:**

1

---

> ### Author Response · Authors · 2021-08-07
> **Thanks for your review comments.**
>
> First of all, we appreciate your efforts in reading our paper and letting us know your comments. We answer your questions as follows.
>
> Invertible neural networks require that the output size is the same as the input size. Our invertible generator’s latent vector input size should be the same as the output hidden vector size. The main purpose of adding the autoencoder is to match the input and the output sizes of the invertible generator.
>
> With the Hutchinson estimator, we do not need to explicitly calculate the Jacobian determinant. Calculating the exact Jacobian determinant is not feasible for large neural networks. So we adopted the Hutchinson estimator which is an unbiased statistical method.
>
> We will also improve our theoretical discussions regarding the complexity and suitability of our method.

---

### Public Comment · Authors · 2022-10-08
**Code is Availiable**

We finalized our code license and our code is available at https://github.com/leejaehoon2016/ITGAN.

Author

---

### Decision · Program_Chairs · 2021-09-27

**Decision:**

Accept (Poster)

**Comment:**

The reviewers have come to an agreement that the paper is both original in content and provides a through experimental development of the technique, with adequate comparisons to strong baselines.

Even when already polished, I would still recommend to try to polish a bit further the narrative in regards to the discussion with RvTau and your reply to 1ZTc. The approach is sophisticated and could benefit from it.

AC.